# Affect Recognition, Theory of Mind, and Empathy in Preschool Children with Externalizing Behavior Problems—A Group Comparison and Developmental Psychological Consideration

**DOI:** 10.3390/children10091455

**Published:** 2023-08-26

**Authors:** Laura M. Watrin-Avino, Franziska J. Forbes, Martin C. Buchwald, Katja Dittrich, Christoph U. Correll, Felix Bermpohl, Katja Bödeker

**Affiliations:** 1Department of Child and Adolescent Psychiatry, Charité Campus Virchow, Charité—Universitätsmedizin Berlin, 13353 Berlin, Germany; f.forbes@outlook.de (F.J.F.); martin.buchwald@charite.de (M.C.B.); katja.dittrich@outlook.com (K.D.); christoph.correll@charite.de (C.U.C.); katja.boedeker@charite.de (K.B.); 2The Zucker Hillside Hospital, Department of Psychiatry, Northwell Health, Glen Oaks, NY 11004, USA; 3Donald and Barbara Zucker School of Medicine at Hofstra/Northwell, Department of Psychiatry and Molecular Medicine, Hofstra University, Hempstead, NY 11549, USA; 4The Feinstein Institutes for Medical Research, Center for Psychiatric Neuroscience, Northwell Health, New Hyde Park, NY 11030, USA; 5DZPG, German Center for Mental Health, Partner Site Berlin, 10785 Berlin, Germany; 6Department of Psychiatry and Neuroscience, Charité Campus Mitte, Charité—University Medicine Berlin, 10117 Berlin, Germany; felix.bermpohl@charite.de

**Keywords:** preschool, externalizing behavior problems, social cognition, affect recognition, theory of mind, empathy

## Abstract

Preschool mental disorders are often associated with significant interpersonal problems, related to impaired affect recognition, theory of mind (ToM), and empathy. To date, these skills have not been studied together in preschoolers with externalizing behavior problems (EBPs). The aim of the present study was to investigate whether and to what extent preschool children with EBPs show impairments in affect recognition, ToM, and empathy. Preschoolers with EBPs, defined by current psychiatric treatment and T-scores ≥ 60 on the externalizing problem scale of the Child Behavior Checklist (CBCL/1½-5 or 6-18R) were compared to non-clinical controls (HCs), defined by no past and no current psychiatric treatment and T-scores < 60 on all CBCL broad-band scales. Groups were compared on affect recognition (NEuroPSYchological Assessment-II), affective ToM (Test of Emotion Comprehension), cognitive ToM (Extended Theory-of-Mind Scale), parent-reported emotional contagion, attention to others’ feelings, and prosocial action (Empathy Questionnaire), IQ and language (Wechsler Preschool and Primary Scale of Intelligence-III Matrices, Active and Passive Vocabulary test), controlling for age, sex, and language abilities. Compared to 27 HCs, 22 preschoolers with EBPs (total sample mean_age_ = 5.6 years +/− 0.8 years, range= 4.2–6.9 years, males 67%) had significantly greater impairments in cognitive ToM (*p* = 0.0002, η^2^ = 0.269), attention to others’ feelings (*p* = 0.0006, η^2^ = 0.236), and prosocial action (*p* = 0.0007, η^2^ = 0.232), each representing strong effect sizes. EBPs were significantly related to cognitive domains, like prosocial action (*r* = −0.490), cognitive ToM (*r* = −0.441), and attention to others’ feelings (*r* = −0.316), but not to affective domains of social cognition. Social cognitive development may be impaired as early as preschool age and should be promoted before the child starts school.

## 1. Introduction

Although epidemiological studies remain sparse, the prevalence of infant mental disorders as per the Diagnostic and Statistical Manual of Mental Disorders (DSM-IV) [1] is consistently estimated to range from 14 to 26.4% [2,3]. In child and adolescent psychiatry, symptoms are classified as either internalizing (e.g., anxiety, depression) or externalizing (e.g., aggression, defiance), although there is considerable and significant co-morbidity [4,5]. Parents and caregivers of young children frequently report externalizing behavior problems (EBPs), such as overactivity, inattention, poor impulse control, and oppositionality, all of which are features of attention-deficit/hyperactivity disorder (ADHD), oppositional defiant disorder (ODD), and conduct disorder (CD) [6]. Externalizing behavior, even as part of normal development, may emerge in infancy and decline with transition to school. As cognitive, language, and executive skills mature, children are increasingly able to cope with developmental tasks and demands and outgrow externalizing behaviors [7,8]. While symptoms are transient in some children, others show persistent impairment [9]. Although in early childhood, behavioral, developmental, and relational problems are closely linked, the association between EBPs and social cognitive skills remains largely unexplored empirically [10,11].

Social cognition has become an important area of research and is considered a potential marker for interpersonal problems across a range of developmental and psychiatric conditions. Social cognition refers to a set of interactive mental processes that help us understand how we and others think and feel [12,13]. Affect recognition is the ability to recognize emotional states in the face or voice of others and precedes the development of theory of mind (ToM). ToM is based on inferring mental states to ourselves and others, including desires, knowledge, beliefs, intentions, and affects [14,15]. A distinction is made between affective and cognitive ToM, depending on whether a person’s affective or mental state is inferred [16,17]. The development of ToM progresses between the 4th and 5th years of life. Language plays an important role in the transition from an initially intuitive to a more reflective and explicit awareness [18]. Empathy is a critical skill we use to regulate our social behavior and relationships. The ability to intuitively imitate and match the emotional expression of others, known as emotional contagion, is considered innate. Children must learn first to regulate their affective arousal and distinguish it from that of others. Only when the child experiences themselves as distinct from the other can they direct their attention to the emotional states of the other and act empathically through prosocial action [19]. ToM and empathy underlie our moral understanding, moral reasoning, and social behavior. From infancy through the early school years, social cognitive skills become increasingly refined, constituting the child’s social competence and adaptability. Social cognitive skills impact a subject’s current and later academic and social success, well-being, and mental health [20,21]. Conversely, a lack of social understanding can lead to interpersonal problems and increase vulnerability to psychopathology [22]. For example, correlations between moral reasoning toward self and others, on the one hand, and between moral reasoning, the expression of empathy, antisocial behavior, criminality, and conflict with the law, on the other, have been reported in the literature for young adults [23]. Blair and Blair (2009) specified the mediating role of emotional dysfunction as decreased feelings of guilt, empathy, and attachment to significant others in antisocial behavior and social dropout [24]. 

Existing studies have addressed different components of social cognition, age groups, and disorders with little overall consistency in definition, operationalization, and measurement, making it difficult to compare the available data. A review of 48 studies of social cognition in individuals with ADHD across all age groups found them to perform significantly poorer than healthy controls in affect recognition and ToM. Yet, deficits in ToM were most pronounced, and children were most affected [25]. Within the few available studies on externalizing symptomatology (mostly ODD) and social cognition in preschool children, there is some evidence for impairments in empathy but little for impairments in ToM and affect recognition. In a Turkish community sample of 116 preschool children, assessed for ToM, emotion knowledge, and empathy, only deficits in empathy were significantly associated with disruptive behavior [26]. In a Canadian sample of 74 4–8-year-old children with ODD, assessed for emotion perception and, separately, affective and cognitive empathy (but not for ToM), only a subgroup of 25 boys with high levels of callous-unemotional traits showed deficits in cognitive empathy [27]. Deficits in empathy have been shown to predict later EBPs in infants as young as 3–36 months of age, particularly in boys [28]. In a community sample of 67 infants, ToM, sympathy (e.g., empathy), and ODD were negatively associated [10]. However, other work has not confirmed these findings [22]. In summary, there is some evidence of a possible influence of social cognitive skills on disruptive behaviors in preschool children [29].

In contrast to most studies that focus on single aspects, the present study examined all clinically relevant components of social cognition in an understudied age group of clinically challenging preschool children with EBPs. The question of whether these EBPs in preschool children are associated with impairments in their social understanding was addressed. Based on the previous literature reviewed above, we hypothesized that our clinical sample differs from the control sample in all aspects of social cognitive functioning. Identifying specific profiles in the social cognitive skills of preschool children with EBPs may (1) provide further insight into the underlying pathogenesis of EBPs, (2) infer the causal and temporal relationships between EBPs and impaired social cognitive skills, (3) allow targeting very specific social cognitive skills, and (4) prevent further maladjustment. 

## 2. Materials and Methods

In this cross-sectional observational study, clinically referred children with EBPs were examined on affect recognition, emotion comprehension, cognitive ToM, empathy-related emotional contagion, attention to others’ feelings, and prosocial action and compared to a non-clinical control sample. We hypothesized that subjects of the clinical sample would lag behind those of the control sample in terms of the above skills (as dependent variables), adjusting for age, sex, and language as covariates.

### 2.1. Sample

Children of the clinical sample were recruited at the Department of Child and Adolescent Psychiatry, Charité—Universitätsmedizin Berlin, and by local child psychiatrists and psychotherapists. Children aged 4.0–6.11 years were eligible for the study. Inclusion criteria for the clinical sample were EBPs in need of treatment and T-scores ≥ 60 on the externalizing problem scale of the CBCL/1½-5 or 6-18R [30,31], including hyperactivity, inattention, impulsivity, tantrums, aggression, rule-breaking behavior, and low frustration tolerance. Subjects with autism spectrum disorder, fetal alcohol syndrome, and intelligence impairment (nonverbal IQ ≤ 70) were excluded from the study. Different cut-off values of T ≥ 65 [32] or even T ≥ 70 [33] are reported in the literature. In the present study with preschool children, we set T ≥ 60, following the recommendation of Achenbach (1991), according to whom the most efficient discrimination between deviant and nondeviant children for most groups and regardless of age and sex is 60 > T < 63 (lowest possible number of false positives and false negatives) [34]. Children of the control sample, recruited from cooperating day care centers, were not allowed to be or have been receiving psychiatric treatment and had to have T-scores < 60 on all CBCL broad-band scales. In addition to the information about the child’s psychopathology collected with the CBCL/1½-5 or 6-18R, another questionnaire asked parents about possible diagnoses and any psychiatric/psychotherapeutic treatments the child had received. Ethical approval was granted by the ethics committee of Charité Universitätsmedizin Berlin. The number of the ethics committee approval is Ethics Committee Charité EA02/007/18, date: 10 April 2018. All parents gave written informed consent.

### 2.2. Procedures

Study recruitment and assessment took place from September 2018 to March 2019. A broad test battery of standardized procedures was used to assess preschool intelligence, language, affect recognition, emotion comprehension, cognitive ToM, empathy-related emotional contagion, attention to others’ feelings, and prosocial action. Children of the clinical sample were assessed in the clinical setting; children of the control sample were tested in a low-stimulus setting of a child care center. Assessment lasted approximately 45–60 min with minor individual breaks to help children with concentration problems. Tests were always administered in the same order: 1. Wechsler Preschool and Primary Scale of Intelligence (WPPSI)-III: Active Vocabulary test, Passive Vocabulary test, and Matrices test [35,36]; 2. NEuroPSYchological Assessment (NEPSY)-II subtest Affect Recognition [37]; 3. Extended Theory of Mind scale [17,38]; and 4. Test of Emotion Comprehension (TEC) [39,40].

### 2.3. Assessments

#### 2.3.1. Demographics

Age and sex of the children, as well as the parent’s socioeconomic status (SES), were recorded. The SES index was computed based on the variable that had been used in epidemiological studies of the German public health institute [41]. The SES index is a multi-dimensional aggregate index calculated as a score on a scale ranging from 1.0 to 7.0 for each of the three dimensions: education (operationalized as an individual’s characteristic based on the respondents’ educational and professional qualification), occupation, and income (household characteristics). Since the three subscales are included in the calculation with the same weight, the SES index can assume values between 3.0 and 21.0. The SES index can be included in analyses as a metric variable, or it can be categorized into multiple status groups [41].

#### 2.3.2. Psychiatric Diagnoses

Some children in the clinical sample had already undergone outpatient psychiatric assessment and received an Axis-I clinical diagnosis, while others were still in the diagnostic process and did not yet have psychiatric diagnoses when they were screened as part of the study. For all children, psychotherapeutic or medication interventions had not yet taken place. Diagnoses were assigned by early childhood mental health experts—clinical psychologists or medical doctors—using the International Statistical Classification of Diseases, German Modification (ICD-10-GM). Of the 22 children of the clinical group, 11 children had established or suspected diagnoses of hyperkinetic disorders according to the ICD-10 (9 children F90.0, 2 children F90.1), and 3 children had mixed disorders of conduct and emotions (ICD-10: F92.8). Four children were diagnosed with unspecified behavioral and emotional disorders (ICD-10: F98.9 or F93.9) because of temper tantrums, impulsivity, and low frustration tolerance. For four children of the clinical group, diagnoses were not available due to an ongoing diagnostic process, all four children seeking treatment because of low frustration tolerance and rule-breaking, aggressive, or provocative behavior. Before patients were enrolled in the study, a clinical interview was conducted by the primary clinician. The clinical interview included an overview of the child’s current behavioral problems, a comprehensive psychiatric status, and developmental history. The exclusion of ASD and FASD was based on the information from the clinical interview. However, diagnoses were not considered in the group assignment and statistical analyses.

#### 2.3.3. Child Behavior Checklist (CBCL)

The CBCL is a widely used, standardized parent form for the dimensional screening for psychiatric symptomatology. A total of 113 items of three broad-band categories (internalizing problems, externalizing problems, and total problems) are rated on a three-point Likert scale. In this study, the present form (CBCL/1½-5) and the school-age assessment form (CBCL 6-18R) were used. The CBCL/1½-5 or 6-18R score for externalizing problems was used for group assignment. T-scores are reported because clinical practitioners are more familiar with them, as the vast majority of psychological and psychiatric questionnaires report results in T-scores. A cut-off of T ≥ 60, independent of age and identical for both forms, was considered borderline and the clinical range to exclude individuals with subclinical symptomatology from the non-clinical control sample. The accompanying caregivers, usually the mothers, filled out the form. For the German version, a good to very good reliability of the externalizing problem scale is reported (CBCL/1½-5: Cronbach’s α > 0.92 [42]; CBCL 6-18: Cronbach’s α > 0.88 [31]).

#### 2.3.4. Affect Recognition

Affect recognition was assessed using a subtest of the NEuroPSYchological Assessment-II, an established test battery with 36 subtests covering six domains of neuropsychological development in 3–16-year-olds. The subtest Affect recognition contains nonverbal tasks for recognizing emotions. Children were asked to match expressions of affects in photographs of faces. The number of tasks presented varied from 16 to 25, depending on the child’s age. The NEPSY-II has shown good reliability (*r* > 0.80) [37].

The Affect recognition score was used to test our hypothesis about differences in affect recognition across samples.

#### 2.3.5. ToM

##### Affective ToM

In order to assess affective ToM, the German version of the Test of Emotion Comprehension (TEC) was administered. The scale includes nine components of emotion knowledge: 1. Recognition of emotions, based on facial expressions, 2. External causes of emotions, 3. Desires as causes of emotions, 4. Beliefs as determinants of emotions, 5. Memory and emotional states, 6. Emotion regulation, 7. The ability to hide emotions, 8. The concept of mixed emotions, and 9. The role of morality in emotions, with, in total, 23 items. For each component, 1–4 tasks were presented as stories in a picture book together with drawings, displaying different emotional states of the story’s protagonist. The children received one point for each correctly solved component, with the total score ranging from 0 to 23.

##### Cognitive ToM

Cognitive ToM was assessed using the German version of the Extended Theory-of-Mind Scale (EToM). The EToM is a validated instrument that measures six components of preschoolers’ ToM using one task each: 1. Diverse Desire, 2. Diverse Belief, 3. Knowledge-Access, 4. Contents False Belief, 5. Hidden Emotion, and 6. Sarcasm. A total score (0–6) was calculated from one point per correctly answered task. The Explicit-False-Belief task was not included, as it did not provide any additional information [43,44]. Tasks were presented in ascending order. In previous studies, the EToM has been shown to be a valid instrument for ToM assessment in preschool children.

The TEC and EToM scores were used to test hypotheses about sample differences in emotion comprehension and cognitive ToM.

#### 2.3.6. Empathy

We used the Empathy Questionnaire (EmQue) to assess empathy-related behavior in 1–6-year-old children through parent report on 20 items over the past two months [45]. The selected procedure conceptualizes three factor-analyzed scales as a developmental pathway: from being affected by the emotions of others (emotional contagion) to cognitive awareness of the emotions of others (attention to others’ feelings) to prosocial action. The English version was translated from English into German and back-translated by a professional translator. The final version was approved by the authors. Coefficients for internal consistency were satisfactory for all three scales of the German version: Emotional contagion α = 0.59, Attention to others’ feelings α = 0.81, and Prosocial action α = 0.84.

The EmQue scales were used to test our hypotheses regarding differences in the samples’ emotional contagion, attention to others’ feelings, and prosocial actions.

#### 2.3.7. Intelligence and Language

##### Intelligence

The Wechsler Preschool and Primary Scales of Intelligence-II—German Version (WPPSI-II), adaptation of the Wechsler Preschool and Primary Scales of Intelligence-III (WPPSI-III), is a widely used test procedure for assessing cognitive abilities in children aged 3.0–7.2 years. In addition to the child’s overall intellectual level (i.e., total IQ), verbal and action skills, a processing speed and general language score can be generated. Since intelligence performance was not the focus of this study, only 4 of the 14 subtests were administered: the Matrix test, consisting of 17 tasks in which the child looks at an incomplete matrix and selects the missing part from 4–5 answer choices, was used as a measure of fluid intelligence.

##### Language

The WPPSI-III Passive Vocabulary test measures the ability to understand verbal instructions in 31 tasks by looking at 4 pictures and pointing to the picture named by the test administrator. The Passive Vocabulary test also measures discrimination of auditory and visual information. Responses may be influenced by performances of phonological memory and working memory. In the Active Vocabulary test, the child is asked to name a picture in a stimulus book (26 tasks) in order to assess its linguistic expression, access to long-term memory, and linkage between visual information and language. Both tests, the Passive and Active Vocabulary tests, were combined to form a general language scale as an indicator of a child’s expressive and receptive language development. The subscales of the WPPSI-III showed good reliabilities (0.77 < *r* < 0.88) [35].

### 2.4. Data Analysis

Primary outcomes were affect recognition, emotion comprehension, cognitive ToM, empathy-related emotional contagion, attention to others’ feelings, and prosocial action. To examine the differences between the samples, we used analyses of covariance (ANCOVAs) for each of the domains of social cognition as dependent variables adding covariates to the group comparison, as our samples differed in descriptive aspects (see Table 1). We examined the effect of a nominally scaled independent variable (grouping variable) on measures of social cognition using ANCOVAs. Alternatively, we could have computed a regression analysis to examine the effect of an interval-scaled independent variable on measures of social cognition. However, we found that EBP scores were bimodally distributed (i.e., children had either low scores or very high scores), suggesting that grouping and analysis of variance accounting for covariates would be the more appropriate procedure. Meta-analyses have consistently shown that ToM in preschool children is influenced by their language development status and by age [46,47,48]. A negative correlation was found between empathy and early EBPs, influenced by the child’s sex and age [28]. Thus, we used age, sex, and language as covariates in all analyses comparing the two groups.

We also examined the strength of the linear relationship between the individual components of social cognition and externalizing and internalizing behavior problems pooled across study groups by using Pearson product-moment correlations. Statistical analyses were conducted using the Statistical Package for the Social Sciences (SPSS, Version 27.0) [49] with two-sided tests, α = 0.05, and with Bonferroni control for multiple comparisons.

## 3. Results

### 3.1. Study Sample

Altogether, 23 clinical and 33 non-clinical control children participated in the study. Of these, 7 children were excluded from the analyses (one clinical subject and six controls) for either not participating in the testing or not meeting the CBCL cut-off criteria, leaving 22 patients and 27 control subjects for the analyses (Table 1).

### 3.2. Descriptive Statistics

As specified by our grouping variable, externalizing problem scores were significantly higher in the clinical sample compared to the control sample (*t*(47) = 11.716, *p* < 0.0001). The same was true for internalizing problem scores (*t*(47) = 4.946, *p* < 0.0001) (Table 1).

In addition, significant sample differences existed in the subjects’ sex distribution (χ^2^(1) = 6.566, *p* = 0.0104, φ = 0.366), age (*t*(47) = 2.161, *p* = 0.0359), SES (*t*(24.99) = 2.878, *p* = 0.0081), and active vocabulary (*t*(47) = 3.694, *p* < 0.0006). In receptive vocabulary (*t*(47) = 1.450, *p* = 0.1538) and general IQ (*t*(38.09) = 1.128, *p* = 0.2664), samples did not differ. Additional sample characteristics are shown in Table 1.

### 3.3. Intercorrelation of Variables Examined, Pooled across Study Groups

As Table 2 shows, there were no significant correlations between EBPs and each of the affective social-cognitive domains: affect recognition, emotion comprehension (affective ToM), and empathy-related emotional contagion. In contrast, for the association with EBPs, moderate correlations were found for the cognitive domains, such as empathy-related attention to others’ feelings (*r* = −0.316) and cognitive ToM (*r* = −0.441), as well as for prosocial action (*r* = −0.490) (Table 2).

The results of the ANCOVAs are shown in Table 3.

Affect recognition tasks: No significant group differences were found in measures of affect recognition.ToM tasks: While there were no significant group differences in emotion comprehension (affective ToM), groups differed significantly in cognitive ToM (*F*(1,47) = 16.161, *p* = 0.0002, η^2^ = 0.269).Empathy tasks: There were no group differences in measures of emotional contagion. However, groups differed significantly in attention to others’ feelings (*F*(1,47) = 13.622, *p* = 0.0006, η^2^ = 0.236) and prosocial action (*F*(1,47) = 13.309, *p* = 0.0007, η^2^ = 0.232).

As shown in Table 3, with a relatively small sample size, a power [50] well above 0.8 could be achieved for cognitive theory of mind (power 1 − ß = 0.97). However, the power of two empathy variables fell below the desired level of 1 − ß = 0.8 (attention to others’ feelings: 0.51, prosocial action: 0.67). Nevertheless, these variables still show high significance. For the other variables with small effects (affective theory mind, affect recognition) or no appreciable effects (emotion contagion), a larger sample size would have been required.

However, the results show that the examined components of social cognition significantly contribute with different strengths to the discrimination between the examined samples.

## 4. Discussion

In the present study, a systematic investigation of all relevant aspects of social cognition was conducted, using established instruments in a clinically prevalent but scientifically understudied age and symptom group. Data on affect recognition, emotion comprehension, cognitive ToM, empathy-related emotional contagion, attention to others’ feelings, and prosocial action were collected as measures of social cognition in preschool children with and without EBPs and compared in terms of their expressions. Control variables were based on the literature and on the samples’ characteristics. The main findings of this study show that preschool children with EBPs differ from non-clinical control children in terms of cognitive ToM, empathy-related attention to others’ feelings, and prosocial action. In contrast, there were no differences in basal affective performances: affect recognition, emotion comprehension (affective ToM), and empathy-related emotional contagion. These significant group differences related to EBPs were observed when adjusting the analyses for age, sex, and language abilities.

One major finding of our work is that preschool children with EBPs show impairments in cognitive ToM when assessed using the Extended Theory-of-Mind Scale. Although few studies distinguish between affective and cognitive ToM in early child development, our results are consistent with the findings of several other studies: Negative correlations between cognitive ToM and EBPs have been found in two-, three-, and four-year-old children [51]. Dinolfo and Malti (2013) found that interpretive understanding (a measure of ToM) negatively predicted ODD in 4–8-year-old children. In contrast to our present findings and to the above-mentioned studies, e.g., Ekerim-Akbulut and colleagues (2019) found no impairments in ToM skills among 3–6-year-old children with disruptive behaviors. These conflicting results may be due to methodological differences. While Ekerim-Akbulut and colleagues used false-belief tasks as their sole measure of ToM, we used a series of tasks that mapped the children’s understanding of six different mental states [17], actually deliberately omitting the false-belief task because it did not provide any additional information in prior studies.

Based on these results, it can be concluded that impairments in cognitive ToM can be detected in preschool children, provided they are assessed with developmentally sensitive instruments. With overall rather contradictory findings, the question of the association between ToM and preschool EBPs cannot yet be answered satisfactorily. However, based on the present study results, it can be assumed that children who have difficulty reading others’ minds are less able to perceive and meet interpersonal expectations and are more likely to exhibit motor agitation and aggressive behavior out of frustration. The more substantiated evidence on the relationship between cognitive ToM and aggression [52], hyperactivity [43], attention problems, and antisocial personality traits [53] in students supports this assumption but necessitates further research, especially on the direction of the relationship and extension into the preschool age group.

The second major finding of the present work documents deficits in specific empathy scales, namely attention to others’ feelings and prosocial action in preschool children with EBPs, but not in their emotional contagion. Our findings are in line with previous studies of empathy deficits in children with EBPs [26,54,55] or with aggression [56,57]. O’Kearney and colleagues (2017) did not find cognitive empathy deficits in preschoolers and students with ODD but did find them in a subgroup of boys with high levels of callous-unemotional traits. Our results are also consistent with Rieffe and colleagues’ (2010) age-dependent conceptualization of empathy development. The authors found that while in a group of 1–5-year-old children, the emotion contagion measure remained stable, attention to others’ feelings and prosocial action improved with age. The authors suggested that emotion contagion develops into an early-formed, enduring construct within the first year of life [58,59,60], whereas attention to others’ feelings and prosocial action do not fully develop until the early school years. As emotion contagion develops in the first year of life, even before the onset of externalizing symptomatology, no impairments were found in the preschool children studied. The deficits in attention to others’ feelings and prosocial action found in our clinical sample can be interpreted as deficient or delayed social cognitive development and, perhaps, the attentional vulnerability components of empathy. Related to our initial questions (1–4), the results shed light on possible pathogenetic factors of EBPs and point to directions for potential interventions and preventive efforts of EBPs in children. Although the cross-sectional design of our study does not allow us to draw conclusions about causal and temporal relationships between social cognition and EBPs, the associations between cognitive ToM and empathy, on the one hand, and between cognitive ToM and EBPs, on the other, suggest that deficits in understanding other people’s beliefs and desires and limitations in empathy may be involved in the development of EBPs. Thus, targeting these very specific social-cognitive skills could be useful for the prevention of EBPs in children at risk for EBPs and for the psychotherapeutic treatment of EBPs once it has developed. Future prospective studies are needed to test the causal and temporal relationships between social cognition and EBPs and the potential benefits of strengthening social cognitive skills for prevention and treatment efforts related to EBPs.

Having found correlations within social cognitive constructs and between these and EBPs, as well as differences between a clinical and a non-clinical sample, the question for future research is what causality underlies this concurrency and how it develops over time. The question of whether externalizing behavior and social-cognitive problems are comorbidities in the strict sense and distinct impairments, and if so, in what direction they influence each other, or whether they are two aspects of a single neurodevelopmental disorder that may share a third common feature (e.g., a genetic component) [61,62,63], requires further investigation.

## 5. Strengths and Limitations

There are several strengths to this study, including the detailed and extensive assessment of relevant aspects of social cognition, using developmentally sensitive, multi-informant established instruments in a clinically prevalent but scientifically understudied age and symptom group, thereby adding to the literature.

There are also several limitations to discuss for the present study: First and foremost is the lack of matching of the groups. Samples differed in terms of the sex, age, and active vocabulary of the subjects. Although we controlled for these variables statistically, using them as covariates in our analyses, we cannot completely rule out an influence of these variables on our main results. Second, the representativeness of results is limited when studying clinical samples. Third, the sample size is relatively small. Therefore, those components of social cognition, such as affect recognition and affective ToM, that were not statistically significantly related to EBPs may have smaller effect sizes that only emerge as significant factors in larger study samples. However, the three main social cognition variables identified in our study, i.e., prosocial action, cognitive ToM, and attention to others’ feelings, had large effect sizes of η^2^ = 0.232–0.269, making a type 1 error unlikely. Fourth, parental assessment of EBPs and clinical observation alone may be biased and may represent an overly broad inclusion criterion that complicates further specification of the relationship with other outcomes. Fifth, and finally, other child factors (e.g., attention, executive functions) [64] or environmental factors (e.g., quality of early social interaction) [65,66] may influence both the manifestation of social-cognitive deficits and externalizing symptomatology, but these were not measured in our study. Future studies should include these domains to more comprehensively assess the relationship between EBPs and social-cognitive and affective domains in preschool children.

## 6. Conclusions

In conclusion, the present study is one of the first to examine social cognition in preschool children with early externalizing symptomatology. By using a developmentally sensitive, broad test diagnostic battery to obtain multi-informant data to identify early disturbances in social cognition, this study makes a useful contribution to the literature on developmental psychopathology. Previous studies have focused on single mental disorders and single aspects of social cognition. Finally, the distinction between affective and cognitive subcomponents of social cognition may facilitate the development of specific therapeutic approaches. The high clinical relevance of social cognition and its diagnostic procedures in preschool children with EBPs could be advanced further.

## Figures and Tables

**Table 1 children-10-01455-t001:** Demographic and clinical characteristics of the samples.

Sample Characteristics	*n*	Clinical Sample *M* (*SD*)	*n*	Control Sample *M* (*SD*)	*df*	χ^2^ or *t*	*p*-Value
Sex							
Male	19		14		1, 47	6.566	0.0104
Female	3		13				
Age (month)	22	70.14 (11.21)	27	63.96 (8.80)	47	2.161	0.0359
Socioeconomic status (SES)	19	12.46 (3.16)	26	14.74 (1.63)	24.99	2.878	0.0081
CBCL externalizing problems	22	71.36 (7.66)	27	47.37 (6.68)	47	11.716	<0.0001
CBCL internalizing problems	22	60.50 (7.99)	27	46.89 (10.69)	47	4.946	<0.0001
WPPSI-III matrix reasoning (gen. IQ)	22	9.91 (3.25)	27	10.85 (2.43)	38.09	1.128	0.2664
WPPSI-III language	22	19.59 (3.96)	27	22.52 (2.90)	47	2.985	0.0045
WPPSI-III active vocabulary	22	9.36 (2.46)	27	11.52 (1.60)	47	3.694	0.0006
WPPSI-III passive vocabulary	22	10.23 (1.90)	27	11.00 (1.82)	47	1.450	0.1538

Legend A. CBCL: Child Behavior Checklist; gen. IQ: general intelligence quotient; WPPSI: Wechsler Preschool and Primary Intelligence Scale. Means (*M*) are presented with standard deviations (*SD*) in parentheses. χ^2^ for analysis of sex. *t* for all other analyses. WPPSI-III scale value scores: *M* = 10, *SD* = 3; WPPSI-III language scale value score: *M* = 20, *SD* = 6.

**Table 2 children-10-01455-t002:** Intercorrelation of social cognition and behavioral variables, pooled across groups.

Social Cognition and Behavioral Variables	1	2	3	4	5	6	7	8
1 Affect recognition	1	0.048	0.008	0.103	0.121	−0.094	−0.203	−0.333 *
2 Emotion comprehension (affective theory of mind)		1	0.535 **	0.013	0.070	0.119	−0.181	−0.010
3 Cognitive theory of mind			1	0.103	0.301 *	0.476 **	−0.441 **	−0.192
4 Empathy Emotion contagion				1	0.434 **	0.365 **	−0.108	0.105
5 Empathy Attention to others’ feelings					1	0.488 **	−0.316 *	−0.253
6 Empathy Proscocial action						1	−0.490 **	−0.257
7 Externalizing problems							1	0.630 **
8 Internalizing problems								1

Legend B. * *p* < 0.05; ** *p* < 0.01.

**Table 3 children-10-01455-t003:** ANCOVAs with age, sex, and language as covariates.

Domains of Social Cognition	Clinical Sample (*n* = 22) *M* (*SD*)	Control Sample (*n* = 27) *M* (*SD*)	*df*	F	*p*-Value *	Part. η*^2^*	Power 1 − ß
Cognitive theory of mind	3.41 (1.33)	4.74 (1.02)	1, 47	16.161	0.0002	0.269	0.97
Affective theory of mind	5.05 (1.81)	5.74 (1.77)	1, 47	1.849	0.1808	0.040	0.09
Emotion contagion	0.37 (0.30)	0.45 (0.26)	1, 47	0.322	0.5733	0.007	0.05
Attention to others’ feelings	1.31 (0.47)	1.62 (0.31)	1, 47	13.622	0.0006	0.236	0.51
Affect recognition	8.86 (2.80)	10.56 (2.49)	1, 47	1.335	0.2542	0.029	0.31
Prosocial action	0.81 (0.50)	1.21 (0.37)	1, 47	13.309	0.0007	0.232	0.67

Legend C. Means (*M*) are presented with standard deviations (*SD*) in parentheses. * Bonferroni adjusted significance level: *p* = 0.05/6 = 0.008. Part. η^2^ = 0.01 small effect, η^2^ = 0.06 moderate effect, η^2^ = 0.14 strong effect.

## Data Availability

The data supporting the results of the study are available upon request from the corresponding author, L.M.W.A.

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
