# Peer review of "Affect Recognition, Theory of Mind, and Empathy in Preschool Children with Externalizing Behavior Problems—A Group Comparison and Developmental Psychological Consideration"

_children, 2023, doi:10.3390/children10091455_

Round 1

Reviewer 1 Report

This study explores an interesting and largely neglected topic in children's psychiatry and clinical psychology, i.e., the impact of social cognitive functioning on externalizing behavior problems. The manuscript contains several problems that should be addressed, both at the structural and formal levels; please see below:

-Abstract- What was the objective of the study?

-line 104- „identifying specific profiles” would sound better; „social cognitive skills” is also preferable;

-lines 104-107- All four suggested reasons for conducting this study should be addressed in the „Discussion” or „Conclusion” sections. How do the current study's results contribute to the identification of specific profiles? Or, at least, these reasons might be formulated as future directions for research if no connection between them and  the results of this study can be supported;

-line 119- Can the Authors explain why the cut-off value of CBCL for a clinical sample was considered t60? Because cut-off scores of 65 or even 70 have also been reported in the literature (https://www.ncbi.nlm.nih.gov/pmc/articles/PMC7031768/; https://www.ncbi.nlm.nih.gov/pmc/articles/PMC3059383/);

-lines116-127: Did all children in the clinical sample have to present with a psychiatric diagnosis in order to be considered eligible for this study? What was the distribution of the diagnoses in this sample? EBP in itself is not a diagnosis; please clarify. What criteria were used for excluding ASD and fetal alcohol syndrome? Children in the control group were not allowed to receive treatment, but what about the subjects in the study group? Did they receive pharmacological treatment, psychotherapy, or both? Do the Authors consider that psychotherapy or medication can represent confounding variables? For example, psychotherapy focused on training communicational skills, or medication that decreases anxiety or depression?

-lines 143-145: Only one reference was presented for the SES index; since this is not a widely used instrument, please consider describing its characteristics, and how the values from 3 to 21 are calculated;

-line 207: „3;0-7;2” is not a customary way to express an age interval; maybe consider using commas instead of semicolons?

-line 213: „language” requires a capital letter in this context;

-line 237- please add the SPSS version here;

-line 252- „subjekt” is a typo;

-line 253- What do the Authors mean by the fact that „diagnoses were not included in the group assignment”? Were these variables not included in the statistical analysis? Were they not determined at the initial visit? Each subject had one disorder, or were subjects with dual diagnosis also?

-line 254: How were the healthy controls screened for psychiatric disorders, current or past (e.g., clinical examination, reports from GPs, a panel of structured evaluation methods, or only CBCL scores)?

-line 261- „actice” is a typo;

-lines 377-386: I would suggest that „Conclusions” may be better represented as a distinct chapter than a paragraph in the „Strengths and limitations” section.

Minor editing of English language is needed.

Author Response

RE: Revision of Manuscript ID children-2562131 (Type of manuscript: Article)

Title: “Affect recognition, theory of mind, and empathy in preschool children with externalizing behavior problems. A group comparison and developmental psychological consideration”; Authors: Laura Maria Watrin-Avino, Franziska Johanne Forbes, Martin Christoph Buchwald, Katja Dittrich, Christoph U. Correll, Felix Bermpohl, Katja Bödeker

Dear Dr. Bolten,

thank you for providing us with the opportunity to revise and resubmit the above manuscript for the Special Issue entitled “Behavioral and Emotional Problems in Early Childhood: Prediction, Prevention and Treatment”.

We appreciate the excellent reviews that highlighted a number of important strengths and weaknesses of the manuscript; the revised manuscript reflects how helpful these comments were. We have addressed the reviewers’ comments in an itemized fashion as described below and have transferred all changes into the manuscript, marked in color.

We hope that the revised manuscript is now acceptable for publication in “children” and look forward to hearing from you.

Sincerely,

Laura M. Watrin-Avino, on behalf of all authors

Reviewer #1

This study explores an interesting and largely neglected topic in children's psychiatry and clinical psychology, i.e., the impact of social cognitive functioning on externalizing behavior problems. The manuscript contains several problems that should be addressed, both at the structural and formal levels; please see below:

Comment: Abstract- What was the objective of the study?

Response: Thank you for pointing out the need to state the objective of the study more clearly, as done by the following addition - line 20-22 - (added text underlined):

“The aim of the present study was to investigate whether and to what extent preschool children with EBP show impairments in affect recognition, ToM, and empathy.“

Comment: -line 104- „identifying specific profiles” would sound better; „social cognitive skills” is also preferable;

Response: Thank you for the suggested language improvements, which we have gladly adopted. The sentence now reads – line 112 -:

“Identifying specific profiles in the social cognitive skills (…)”

Comment: -lines 104-107- All four suggested reasons for conducting this study should be addressed in the „Discussion” or „Conclusion” sections. How do the current study's results contribute to the identification of specific profiles? Or, at least, these reasons might be formulated as future directions for research if no connection between them and  the results of this study can be supported;

Response: Thank you for making this important point. We agree and have revisited our introductory questions in the discussion to increase the stringency of the paper. Please find our related edits - line 409-421.

"Related to our initial questions (1-4), the results shed light on possible pathogenetic factors of EBP and point to directions for potential interventions and preventive efforts of EBP in children. Although the cross-sectional design of our study does not allow to draw conclusions about causal and temporal relationships between social cognition and EBP, the associations between cognitive ToM and empathy, on the one hand, and between cognitive ToM and EBP, on the other, suggest that deficits in understanding other people's beliefs and desires and limitations in empathy may be involved in the development of EBP. Thus, targeting these very specific social-cognitive skills could be useful for the prevention of EBP in children at risk for EBP and for the psychotherapeutic treatment of EBP once it has developed. Future prospective studies are needed to test the causal and temporal relationships between social cognition and EBP and the potential benefits of strengthening social cognitive skills for prevention and treatment efforts related to EBP."

Comment: -line 119- Can the Authors explain why the cut-off value of CBCL for a clinical sample was considered t≥60? Because cut-off scores of 65 or even 70 have also been reported in the literature (https://www.ncbi.nlm.nih.gov/pmc/articles/PMC7031768/; https://www.ncbi.nlm.nih.gov/pmc/articles/PMC3059383/);

Response: Thank you for pointing out that different cut-off values have been reported in the literature. We are happy to address this in line 131-136.

“Different cut-off values of T > 65 [32] or even T > 70 [33] have been reported in the literature. In the present study with preschool children, we chose T >/= 60 as the cut-off, following the recommendation of Achenbach (1991), according to whom the most efficient discrimination between deviant and nondeviant children for most groups and regardless of age and sex is 60 > T < 63 (lowest possible number of false positive and false negative rates) [34].”

[32] Levy, S.E.; Rescorla, L.A.; Chittams, J.L.; Kral, T.J.; Moody, E.J.; Pandey, J.; Pinto-Martin, J.A.; Pomykacz, A.T.; Ramirez, A.; Reyes, N.; et al. ASD Screening with the Child Behavior Checklist/1.5-5 in the Study to Explore Early Development. J. Autism Dev. Disord. 2019, 49, 2348–2357, doi:10.1007/s10803-019-03895-4.

[33] Halperin, J.M.; Rucklidge, J.J.; Powers, R.L.; Miller, C.J.; Newcorn, J.H. Childhood CBCL Bipolar Profile and Adolescent/Young Adult Personality Disorders: A 9-Year Follow-Up. J. Affect. Disord. 2011, 130, 155–161, doi:10.1016/j.jad.2010.10.019.

[34] Achenbach, T.M. Manual for the Child Behavior Checklist/4-18 and 1991 Profile; Dept. of Psychiatry, University of Vermont: Burlington, VT, 1991; ISBN 978-0-938565-08-6.

Comment: -lines 116-127: Did all children in the clinical sample have to present with a psychiatric diagnosis in order to be considered eligible for this study? What was the distribution of the diagnoses in this sample? EBP in itself is not a diagnosis; please clarify. What criteria were used for excluding ASD and fetal alcohol syndrome? Children in the control group were not allowed to receive treatment, but what about the subjects in the study group? Did they receive pharmacological treatment, psychotherapy, or both? Do the Authors consider that psychotherapy or medication can represent confounding variables? For example, psychotherapy focused on training communicational skills, or medication that decreases anxiety or depression?

Response: Thank you for all the important questions about the more specific characteristics of our sample. We originally had chosen not to report the clinical diagnoses because we did not use structured interviews in our study, which we mentioned under limitations. We decided to use the CBCL as a measure of the children's externalizing behavioral problems and to define the symptom cluster of EBP. To clarify this point, we added a section - line 170-189 -.

Psychiatric diagnoses

“Some children in the clinical sample had already undergone outpatient psychiatric assessment and received an Axis-I clinical diagnosis, while others were still in the di-agnostic process and did not yet have psychiatric diagnoses when they were screened as part of the study. For all children, psychotherapeutic or medication interventions had not yet taken place. Diagnoses were assigned by early childhood mental health experts - clinical psychologists or medical doctors - using the International Statistical Classification of Diseases, German Modification (ICD-10-GM). Of the 22 children of the clinical group, 11 children had established or suspected diagnoses of hyperkinetic disorders according to the ICD-10 (nine children F90.0, two children F90.1), three children had mixed disorders of conduct and emotions (ICD-10: F 92.8). Four children were diagnosed with unspecified behavioral and emotional disorders (ICD 10: F98.9 or F93.9) because of temper tantrums, impulsivity and low frustration tolerance. For five children of the clinical group, diagnoses were not available due to an ongoing diagnostic process, all five children seeking treatment because of low frustration tolerance, rule-breaking, aggressive or provocative behavior. Before patients were enrolled in the study, a clinical interview was conducted by the primary clinician. The clinical interview included an overview of the child's current behavioral problems, a comprehensive psychiatric status and developmental history. The exclusion of ASD and FASD was based on the information from the clinical interview. However, diagnoses were not considered in the group assignment and statistical analyses.”

Comment: -lines 143-145: Only one reference was presented for the SES index; since this is not a widely used instrument, please consider describing its characteristics, and how the values from 3 to 21 are calculated;

Response: You suggested that we explain the SES survey instrument, developed in Germany in more detail to the international readership, a suggestion we were happy to follow. We have included a more detailed description - lines 161 to 169 -.

“The SES index was computed based on the variable that had been used in epidemiological studies of the German public health institute (Lampert et al., 2013).  The SES index is a multi-dimensional aggregate index calculated as a score on a scale ranging from 1.0 to 7.0 for each of the three dimensions: education (operationalized as an individual’s characteristic based on the respondents' educational and professional qualification), occupation, and income (household characteristics). Since the three subscales are included in the calculation with the same weight, the SES index can assume values between 3.0 and 21.0. The SES index can be included in analyses as a metric variable, or it can be categorized into multiple status groups (Lampert et al., 2013).”

Comment: -line 207: „3;0-7;2” is not a customary way to express an age interval; maybe consider using commas instead of semicolons?

Response: Thank you for bringing the discrepancy in the presentation of age to our attention. We have changed this in the article – lines 250, 31 -.

3.0-7.2 years

Comment: -line 213: „language” requires a capital letter in this context;

Response: Sorry for the mistake, this was corrected - line 259 -.

Comment: -line 237- please add the SPSS version here;

Response: Thank you very much for pointing this out. We have added the SPSS version 27.0 - line 287/288.

Comment: -line 252- „subjekt” is a typo;

Response: Thank you for the very attentive reading. We ultimately deleted this section on diagnoses from the results section.

Comment: -line 253- What do the Authors mean by the fact that „diagnoses were not included in the group assignment”? Were these variables not included in the statistical analysis? Were they not determined at the initial visit? Each subject had one disorder, or were subjects with dual diagnosis also?

Response: We have inserted all information about diagnoses – line 170-189 -. Since diagnoses were not part of the results, we deleted this part.

Comment: -line 254: How were the healthy controls screened for psychiatric disorders, current or past (e.g., clinical examination, reports from GPs, a panel of structured evaluation methods, or only CBCL scores)?

Response: Thank you for asking for this clarification that we have added - line 138-141-.

“In addition to the information about the child's psychopathology collected with the CBCL/1½-5 or 6-18R, another questionnaire asked parents about possible diagnoses and any psychiatric/psychotherapeutic treatments the child had received.”

Comment: -line 261- „actice” is a typo;

Response: Thank you for the very attentive reading. We have changed the typo - line 314 -.

Comment: -lines 377-386: I would suggest that „Conclusions” may be better represented as a distinct chapter than a paragraph in the „Strengths and limitations” section.

Response: Thank you very much, we gladly followed this suggestion - line 454 -.

Comments on the Quality of English Language: Minor editing of English language is needed.

Reviewer 2 Report

This is an interesting article, but there are some major methodological issues:

The sample size is small. Researchers should justify the sample size and rerun analyses with larger samples as conclusions can not be generalized.

Why were these tests chosen? Please justify why these instead of other similar tests.

Please describe the presented tables in the results section.

Please mention the number-protocol of the decision coming from the ethics committee in the text.

Why t scores were used instead of z-scores? Are z-scores better in the analyses?

In addition to that, the variables mentioned (affect recognition, theory of mind, and empathy) have also been linked to criminal behavior in individuals with and without Intellectual Disabilities as those included in this sample of children (please see

Giannouli, V. (2016). Crime and legal issues among intellectually disabled individuals. In Handbook of research on diagnosing, treating, and managing intellectual disabilities (pp. 346-369). IGI Global.)   In the statistics which version of SPSS did the authors use?

Εnglish language editing is necessary.

Author Response

please see the attachment, thank you. 

Round 2

Reviewer 1 Report

The manuscript significantly improved. Congratulations to the Authors for their hard work.